# The Effect of Sodium Butyrate on Adventitious Shoot Formation Varies among the Plant Species and the Explant Types

**DOI:** 10.3390/ijms21228451

**Published:** 2020-11-10

**Authors:** Myoung Hui Lee, Jiyoung Lee, Seung Hee Choi, Eun Yee Jie, Jae Cheol Jeong, Cha Young Kim, Suk Weon Kim

**Affiliations:** Biological Resource Center, Korea Research Institute of Bioscience and Biotechnology (KRIBB), Jeongeup 56212, Korea; mhlee17@kribb.re.kr (M.H.L.); jiyoung1@kribb.re.kr (J.L.); csh@kribb.re.kr (S.H.C.); jeannie@kribb.re.kr (E.Y.J.); jcjeong@kribb.re.kr (J.C.J.); kimcy@kribb.re.kr (C.Y.K.)

**Keywords:** histone deacetylase inhibitor, sodium butyrate (NaB), histone acetylation, in vitro tissue culture, protoplasts, gene expression

## Abstract

Histone acetylation plays an important role in plant growth and development. Here, we investigated the effect of sodium butyrate (NaB), a histone deacetylase inhibitor, on adventitious shoot formation from protoplast-derived calli and cotyledon explants of tobacco (*Nicotiana benthamiana*) and tomato (*Solanum lycopersicum*). The frequency of adventitious shoot formation from protoplast-derived calli was higher in shoot induction medium (SIM) containing NaB than in the control. However, the frequency of adventitious shoot formation from cotyledon explants of tobacco under the 0.1 mM NaB treatment was similar to that in the control, but it decreased with increasing NaB concentration. Unlike in tobacco, NaB decreased adventitious shoot formation in tomato explants in a concentration-dependent manner, but it did not have any effect on adventitious shoot formation in calli. NaB inhibited or delayed the expression of D-type cyclin (*CYCD3-1*) and shoot-regeneration regulatory gene *WUSCHEL* (*WUS*) in cotyledon explants of tobacco and tomato. However, compared to that in control SIM, the expression of *WUS* was promoted more rapidly in tobacco calli cultured in NaB-containing SIM, but the expression of *CYCD3-1* was inhibited. In conclusion, the effect of NaB on adventitious shoot formation and expression of *CYCD3-1* and *WUS* genes depended on the plant species and whether the effects were tested on explants or protoplast-derived calli.

## 1. Introduction

The ability of plants to dedifferentiate and regenerate cells from differentiated somatic tissues in plants has served as an important tool for producing fully functional plantlets from plant explants or cells. The regenerative capacity of plant cells can be enhanced in vitro in nutrient media supplemented with plant hormones [1,2], and in particular, the ratio of auxin to cytokinin plays an important role in the determination of shoot, root, or callus differentiation [1,3]. Callus formation from explants can be induced on auxin-rich callus-inducing medium (CIM) and shoot regeneration from callus cells can be induced on cytokine-rich shoot induction medium (SIM) [4].

In *Arabidopsis*, the genes such as *CUP*-*SHAPED COTYLEDON* (*CUC*), *SHOOT MERISTEMLESS* (*STM*), *WUSCHEL* (*WUS*), and many others are required for shoot apical meristem (SAM) formation during embryogenesis for SAM maintenance during post-embryonic development [5,6,7,8]. *CUC1* and *CUC2,* encoding a pair of paralogous NAC transcription factors, are required for shoot meristem initiation through the promotion of *STM* expression [9]. *STM* promotes cell division and inhibits cell differentiation in the SAM [10]. *WUS*, a homeodomain transcription factor, is essential for reprogramming during de novo shoot regeneration [11]. One study showed that a *wus* mutant failed to regenerate the shoots, whereas the overexpression of *WUS* led to shoot regeneration on a hormone-free medium [12]. Type-B Arabidopsis response regulators (ARRs) play an important role in the regulation of auxin levels by cytokinin signaling [13], which leads to cell-cycle re-entry via the up-regulation of D-type cyclin *CYCD3;1* [14]. *CYCD3;1* functions as a downstream effector of cytokinin signaling, and it is involved in the control of the cell cycle at the G1/S transition [15].

Epigenetic modifications are also important for plant differentiation and organogenesis. In *Arabidopsis*, dynamic regulation of gene expression at the chromatin level plays an important role in translating developmental and environmental signals that regulate cellular totipotency [16]. This regulation is primarily mediated by chromatin remodeling and histone modification [17,18]. Histone acetylation and deacetylation are regulated by histone acetyltransferase (HAT) and histone deacetylases (HDAC), respectively [19]. HDAC are a class of enzymes responsible for removing acetyl groups from the lysine residue of both histone and nonhistone proteins [20]. Histone deacetylase inhibitors (HDI) are a group of agents that inhibit histone deacetylase enzymes. HDAC inhibition alters the balance for histone acetylation, chromatin relaxation, and gene expression [21,22]. Trichostatin A (TSA) is a member of a larger class of HDIs which selectively inhibits the class I and II mammalian HDAC enzyme families [23]. In *Brassica napus*, TSA treatment increased histone H3 and H4 acetylation of the *WUS* genomic region [24] and upregulated a few members of cell cycle-related genes in the male gametophyte [25]. In addition, cell cycle-related genes were up- and down-regulated by many HDI treatments [26]. Sodium butyrate (NaB) is a short-chain fatty acid and which functions as an HDI [27,28]. NaB induces growth arrest, differentiation, and apoptosis in cancer cells, primarily through its effects on HDAC activity [29].

HDIs have been reported to affect a variety of plant physiological responses in a concentration-dependent manner. TSA has been reported to induce various plant physiological responses, including inhibition of embryo maturation, inhibition of seed germination and seedling development, promotion of embryogenic cell proliferation, and inhibition of somatic embryogenesis, depending on the treatment concentration [25,30,31,32,33,34]. Similarly, NaB has been reported to induce a wide variety of plant physiological responses, such as repression of seed germination and seedling development, cell division in explants, and inhibition of early seedling development and root elongation, which also depend on the treatment concentration [34,35,36]. In addition, Furuta et al. (2011) reported that TSA induces similar effects as cytokinin on callus formation in *Arabidopsis* [37].

Despite extensive studies on the role of histone modifications in eukaryotic gene regulation, the effects of NaB on the adventitious shoot formation are not well understood. Therefore, in the present study, we investigated the effects of NaB on adventitious shoot formation in tomato and tobacco cotyledon explants as well as in their protoplast-derived-calli. We also examined the differential gene expression patterns of *CYCD3-1* and shoot regeneration regulatory gene *WUS* from cotyledon explants and protoplast-derived-calli of tobacco and tomato depending on the NaB concentration.

## 2. Results

### 2.1. Sodium Butyrate Significantly Enhanced the Growth and Adventitious Shoot Formation of Tobacco Calli

We investigated the effect of NaB on callus growth and adventitious shoot formation in the callus induction medium (CIM). The results showed that callus size increased under the NaB treatment in a concentration-dependent manner: it increased significantly by approximately 1.38- and 1.71-fold under 0.5 mM and 1.0 mM NaB treatments, respectively, compared to that in the control (Appendix A).

We also examined the effect of NaB on callus growth and adventitious shoot formation in the shoot induction medium (SIM) (Figure 1). Similar to the previous experiment, callus size was significantly increased in the NaB-containing media compared to that in the control media, and the increase was in a concentration-dependent manner (Figure 1A,B). Callus size was calculated every week from the third to the sixth week of the culture. Compared to that at the beginning of culture, the callus size was increased by approximately 1.6-, 2.1-, and 1.86-fold under the 0.1, 0.5, and 1.0 mM NaB treatments, respectively, after 3 weeks of culture, whereas after 6 weeks of culture, this increase was 2.5-, 2.8-, and 2.5-fold, respectively (Figure 1B). In addition, the frequency of adventitious shoot formation from tobacco calli was examined after 6–8 weeks of incubation in SIM. After 6 weeks of incubation in SIM containing 0, 0.1, 0.5, and 1.0 mM of NaB, the frequency of adventitious shoot formation was 5%, 15%, 86%, and 86%, respectively; thus, it was significantly higher in the SIM containing 0.5 and 1.0 mM of NaB than in the SIM containing 0 and 0.1 mM of NaB (Figure 1C). These results indicated that NaB stimulated callus growth in both CIM and SIM, and stimulated adventitious shoot formation in tobacco calli.

### 2.2. NaB Does Not Have a Positive Role in Enhancing Callus Growth in Tomato

We examined the effect of NaB on callus growth in tomato (Appendix A). Under the treatment with SIM-1, callus growth was slightly decreased in NaB containing media compared to that in the control media (Appendix A). After 6 weeks of incubation in SIM-1, the exact callus size of protoplast-derived calli treated with 0, 0.1, 0.5, and 1.0 mM of NaB decreased by approximately 0.96-, 0.92-, and 0.90-fold, respectively, compared to that in the control (Appendix A). The overall growth effect of NaB on protoplast-derived calli growth was not positive, and its inhibitory effect on callus growth was concentration-dependent. Even when the culture period was extended, the growth inhibition continued. The inhibitory effect of NaB on protoplast-derived calli growth in SIM-2 showed a similar pattern to that in SIM-1 (Appendix A). After 6 weeks of incubation in SIM-2, the callus size of protoplast-derived calli treated with 0, 0.1, 0.5, and 1.0 mM of NaB decreased by approximately 0.96-, 0.86-, and 0.8-fold, respectively, compared to that in the control (Appendix A). These results showed that rather than having a stimulating effect, NaB has an inhibitory effect on callus growth in tomato, which differed from the results obtained for tobacco calli. Unfortunately, in both SIM-1 and SIM-2, the frequency of adventitious shoot formation in tomato calli was very low, making it difficult to estimate the effects of NaB on shoot formation (Appendix A).

### 2.3. Adventitious Shoot Formation in Cotyledon Explants of Tobacco is Significantly Affected by the NaB Concentration

We already showed that NaB had a significant role in accelerating adventitious shoot formation in tobacco calli (Figure 1C). In order to investigate whether such a growth-promoting effect of NaB exists in other types of tissues, its effect on cotyledon explants was examined in the same manner. Under the 0.1 mM NaB treatment, explant size did not show much difference from that in the control. However, as the NaB concentration increased, the explant size decreased significantly (Figure 2A,B). After 5 weeks of incubation in SIM containing 0.5 and 1.0 mM of NaB, the explant size was reduced by approximately 0.6-, and 0.5-fold, respectively, compared to that in the control (Figure 2B). These results indicated that the increase in cotyledon explant size in tobacco was inhibited by the NaB treatment in a concentration-dependent manner.

New callus formation in tobacco cotyledon explants was increased significantly after 4 weeks of cultivation under both control and 0.1 mM NaB treatment, but this increase was significantly delayed under 0.5 mM and 1.0 mM NaB treatments compared to that in the control, and during 6 weeks of culture, the plants under these two treatments did not recover as much as those in the control. After 4 weeks of treatment, the percentage of shoots with a length of 3 mm or more under 0, 0.1, 0.5, and 1.0 mM of NaB treatments were approximately 11%, 26%, 3.7%, and 5.5%, respectively. However, when this period was prolonged to 6 weeks, the formation of adventitious shoots equal to or longer than 3 mm under 0, 0.1, 0.5, and 1.0 mM treatments was higher than 96%, 99%, 93%, and 75%, respectively (Figure 2C). Under the 1.0 mM NaB treatment, the frequency of adventitious shoot formation reached 78% of that in the control. However, the number of shoots per explant under 0.5 and 1.0 mM NaB treatments significantly decreased to 60% and 30%, respectively, than that in the control (Figure 2D). Interestingly, the overall length of shoots was almost similar under the control and NaB treatments after 6 weeks of incubation (Figure 2E).

These results indicated that a low concentration of NaB (0.1 mM) had similar or slightly positive effects on the formation of adventitious shoots. However, high concentrations of NaB (0.5 and 1.0 mM) negatively affected adventitious shoot formation in tobacco cotyledon explants.

### 2.4. NaB Inhibits New Shoot Formation, but Improves Normal Shoot Growth in Cotyledon Explants of Tomato

The effects of NaB on adventitious shoot formation and development of shoots from cotyledon explants of tomato were examined (Figure 3). We previously reported that the efficiencies of plants to form adventitious shoots changed with seedling age [38]. Therefore, in the present study, we used plant explants of 7 days after germination (DAG7), which had moderate shoot formation efficiency. The overall efficiency of morphologically normal shoot formation (excluding abnormal-shaped shoots, such as those with a single leaf) formation was similar regardless of NaB treatment in SIM after 5 weeks of incubation. Adventitious shoot formation continued throughout the control treatment, whereas in NaB-treated cotyledon explants, it decreased in a concentration-dependent manner (Figure 3A,B). Interestingly, however, the number of abnormal-shaped shoots (such as examples with a single leaf) significantly decreased in NaB-treated cotyledon explants compared to that in the control. The average number of normal-shaped shoots was about 1.5 per explant, and there was no significant difference not only between the control and the NaB treatments but also among the NaB treatments (Figure 3B). The efficiency of adventitious shoot formation was not significantly higher under the NaB treatment compared to that in the control (Figure 3C). In addition, the explant size greatly decreased in NaB containing medium compared to that in the control medium, and the decrease was concentration-dependent (Figure 3D, Appendix A). However, the average shoot length was significantly increased compared to that in control under NaB treatments after 5 weeks of culture (Figure 3E). Under 0.5 and 1.0 mM NaB treatments, shoot length was 1.5 and 1.7 times higher, respectively, than that in the control. These results indicated that NaB had no positive effect on adventitious shoot formation in cotyledon explants of tomato, but it improved normal growth of shoots.

### 2.5. Gene Expression of NbCYCD3-1 and NbWUS in Tobacco Calli after NaB Treatment

Although several studies have reported the change in gene expression profiles by HDIs [39,40,41,42,43,44,45,46,47], the exact effect of HDIs on gene expression has not been clearly elucidated to date. Histone modification is widely associated with the transcriptional regulation of cell cycle genes [48]. Li et al. (2014) reported that a small number of cell cycle-related genes (cyclin D3;3, CYCLIN D1-like gene) were upregulated after TSA treatment [25]. TSA treatment enabled rapid de novo activation of *WUS* expression by cytokinin analog BAP [24]. In addition, the expression of the M/G2-phase marker *CYCB1;1* in callus was higher in *hag1* (histone acetyltransferase) mutant than in wild-type plants, whereas *WUS* expression was dispersed in *hag1* callus cultured on SIM and defected shoot formation [49]. Moreover, Lee et al. (2020) reported that the production of *CYCD3-1* and homeodomain-containing transcription factor *WUS* was highly increased during adventitious shoot formation in tomato explants [38]. In the present study, the expression levels of *CYCD3-1*, and *WUS* genes in the calli and cotyledon explants of tobacco and tomato cultured in SIM containing NaB was investigated. In order to investigate whether NaB affects the gene expression of calli and cotyledon explants, the gene expression of *NbCYCD3-1* and *NbWUS* in relation to the incubation period was investigated (Figure 4). In the control, the transcription level of *NbCYCD3-1* reached its maximum after 1 week of incubation in SIM, but *NbCYCD3-1* gradually decreased with the increase in the culture period. The overall expression pattern of *NbCYCD3-1* was similar under 0.1, 0.5, and 1.0 mM NaB treatments, but compared to that in the control, the expression level of *NbCYCD3-1* decreased significantly with the increase in NaB concentration (Figure 4A). The overall expression of *NbWUS* showed various patterns under different NaB treatments (Figure 4B). In the control, the transcription level of *NbWUS* reached its maximum after 3 weeks of incubation in SIM. Similar to that in the control, the transcription of *NbWUS* at the low concentration of NaB (0.1 mM) increased continuously as the culture period increased. However, after 1 week of culture, the expression level of *NbWUS* from at the concentration of NaB (0.1 mM) increased to almost the same level as that after 3 weeks of culture in the control. At a higher concentration of NaB (0.5 and 1.0 mM), the transcription level of *NbWUS* reached its maximum after 2 weeks of incubation in SIM. However, the expression level of *NbWUS* at higher concentrations of NaB decreased as the culture period increased. The overall expression level of *NbWUS* was higher under the 0.5 mM NaB treatment than under the 1.0 mM NaB treatment (Figure 4B).

These results suggested that the expression of *NbWUS* in tobacco calli was more rapidly induced and then reduced by the NaB treatment during the culture in SIM. Furthermore, these results also indicated that the effect of NaB on the early expression of *WUS* varies depending on the NaB concentration. However, the expression of *NbCYCD3-1* decreased significantly with increasing NaB concentration.

### 2.6. Gene Expression of NbCYCD3-1 and NbWUS in Cotyledon Explants of Tobacco

We further investigated the effects of NaB on the gene expression of *NbCYCD3-1* and *NbWUS* in cotyledon explants of tobacco during incubation in SIM containing NaB (Figure 5). The transcription level of *NbCYCD3-1* continued to increase with the increase in incubation length and reached its maximum at 12 days of incubation in control SIM, whereas the transcription level of *NbCYCD3-1* gradually decreased with NaB treatment in a concentration-dependent manner (Figure 5A). The overall expression level of *NbCYCD3-1* increased in NaB-containing medium, but the time taken for this increase depended on the NaB concentration. Under the 1.0 mM NaB treatment, the expression level did not change until DAT5, and it started to increase at DAT8 (Figure 5A). Under the control and low concentration of NaB (0.1 mM) treatment, the transcription level of *NbWUS* in SIM started to increase at DAT5 and reached its maximum at DAT8, after which it started to decline with the increase in the culture period. However, the transcription level of *NbWUS* started to increase at DAT8 and DAG12 under 0.5 mM and 1.0 mM NaB treatments, respectively, and increased until DAT12 (Figure 5B). These results suggested that the expression of both *NbCYCD3-1* and *NbWUS* in cotyledon explants was negatively affected by the NaB treatment compared to that in the control.

### 2.7. Effect of NaB on the Expression of SlCYCD3-1 and SlWUS Genes in Cotyledon Explants of Tomato

We also examined the effects of NaB on the gene expression of *SlCYCD3-1* and *SlWUS* in cotyledon explants of tomato (Figure 6). In the control, the transcription level of *SlCYCD3-1* in SIM gradually increased with the increase in the culture period, reaching its maximum at 12 days of incubation. Interestingly, however, the transcription level of *SlCYCD3-1* in SIM under the NaB treatments reached its maximum after 5 days of incubation; thereafter, the expression level of *SlCYCD3-1* remained almost the same during the rest of the incubation period (Figure 6A). The overall expression patterns of *SlWUS* in cotyledon explants of tomato showed an increase over time under all NaB treatments (Figure 6B). The transcription level of *SlWUS* in SIM reached its maximum at 12 days of incubation. However, the transcription level of *SlWUS* under the NaB treatments was significantly reduced compared to that under the control treatment, and the transcription levels of *SlWUS* at DAT12 under in 0.1, 0.5, and 1.0 mM NaB treatments were similar to that at DAT5 in the control SIM. In addition, the increase in *WUS* expression was significantly suppressed at DAT5 and DAT8 under 0.5 and 1.0 mM NaB treatments, and this suppression was in a concentration-dependent manner. These results showed that the expression levels of *SlCYCD3-1* and *SlWUS* varied under different NaB treatments in tomato cotyledon explants.

### 2.8. NaB Promotes Histone Acetylation in Tobacco Protoplast-Derived Calli

Modifications of chromatin structure occurred primarily during callus formation in an in vitro tissue culture process [50]. TSA treatment increased histone H3 and H4 acetylation of the *WUS* genomic region [24]. To investigate the effect of NaB on histone acetylation, western blot analysis with H3 acetylation antibody and H3 antibody was carried out in tobacco calli during the culture in SIM containing NaB (Figure 7). After 1 and 2 weeks of incubation in SIM containing NaB, histone H3 acetylation in tobacco calli was significantly increased compared to that in the control, and this increase was in a concentration-dependent manner. These results indicated that histone H3 acetylation was induced by NaB in protoplast-derived calli of tobacco.

## 3. Discussion

Both positive and negative effects of HDAC inhibitors have been reported in several plant species. The reported positive effects of HDIs include increasing salinity stress tolerance in *Arabidopsis* [51,52,53] and cassava [54], improving embryogenic cell proliferation, and doubled haploid production in *Brassica napus,* and improving totipotency and embryogenic growth in *Arabidopsis* [25]. In contrast, several studies have reported negative effects of HDIs, such as arresting embryo maturation in the two conifer species [31] and impairing seed germination and seedling development in several plant species [30,32,34,36].

The effects of TSA on callus formation also vary among the explants from different plant parts in plant tissue culture [37,55]. Lee et al. (2016) showed that treatment with TSA in a callus-induction medium led to defective callus formation [55]; however, callus formation from hypocotyl explants was accelerated in *Arabidopsis* [37]. It has been suggested that chromatin remodeling and histone deacetylation in calli or explants are closely related to callus growth [37,55]. In the present study, we primarily revealed that NaB stimulated callus growth and adventitious shoot formation in protoplast-derived calli of tobacco. However, the addition of NaB decreased adventitious shoot formation in the cotyledon explants of tobacco and tomato in a concentration-dependent manner. Furthermore, NaB did not have a positive role in enhancing callus growth and adventitious shoot formation in tomato callus. These results indicated that NaB may have varying effects on callus growth and adventitious shoot formation depending on the plant species and NaB concentrations. Callus growth in tomato was slightly reduced after NaB treatment in SIM in a concentration-dependent manner. However, under this condition, the effect of NaB on adventitious shoot formation was not observed.

Almost all HDIs induce cell-cycle arrest, differentiation, or apoptosis in vitro [56,57]. In addition, cell cycle-related genes were up- and down-regulated by HDI treatment [26]. In the present study, NaB treatment inhibited adventitious shoot formation in both tobacco and tomato explants, but the expression patterns of *CYCD3-1* differed between the plant explants of the two species. In tobacco explants, the expression level of *NbCYCD3-1* was suppressed and significantly delayed by NaB in a concentration-dependent manner. However, in tomato explants, the expression of *SlCYCD3-1* was significantly increased by NaB compared to that in the control at DAT5, after which the level of expression was maintained at high levels by NaB treatment at DAT8 and DAT12. However, the expression of a shoot-regeneration regulatory gene *WUS* was significantly delayed by NaB in both tomato and tobacco cotyledon explants in a concentration-dependent manner. However, *WUS* expression differed between tomato and tobacco explants. In tomato explants, its expression was still increased at DAT12 under the control and NaB treatment, whereas in tobacco, maximal transcript accumulation was observed in the culture under the control and 0.1 mM NaB treatment at DAT8; thereafter, it decreased at DAT12. In addition, *NbWUS* expression was the highest at DAT12 in the 0.5 mM NaB treatment. Moreover, under the 1.0 mM NaB treatment, its expression started to increase at DAT12. It can probably be assumed that in the control, its expression increases and then decreases maximally in the period of DAT5 to DAT8. These results suggested that there was a difference in the expression cycle of *WUS* between tomato and tobacco explants.

The expression patterns of *CYCD3-1* and *WUS* significantly differed depending on the culture time of NaB as well as on whether we used calli or explants. Gao et al. (2019) reported that the gene expression of calli from different explants clustered together [58]. However, in tobacco, the origin of protoplast-derived calli and explants is the same as that of cotyledons, but the expression patterns of cotyledon explants and calli are different. In the present study, it was difficult to accurately compare the gene expression patterns in calli and explants because the treatment time of NaB differed between them. The expression of *NbCYCD3-1* in the callus was the highest in the control after 1 week of incubation, and it gradually decreased with the increase in incubation time; however, the expression of *NbCYCD3-1* in explants continued to increase until day 12 of incubation. In addition, the expression of *NbWUS* was more rapidly induced and reduced by NaB in explants than in calli. The NaB treatment delayed the expression of *NbCYCD3-1* and *NbWUS* in tobacco explants; however, it promoted the expression of *NbWUS* but the expression of *NbCYCD3-1* decreased significantly with increasing NaB concentration in calli derived from tobacco protoplasts. It is assumed that the reasons for the different expression patterns of cotyledon explants and calli are the different stages of shoot formation.

Global gene expression profiles indicated that only a small proportion of genes in mammalian and plant cells respond to HDIs [25,39,40,41,42,43,44,45,46,47]. In addition, previous studies have reported that the incubation time and concentration of HDI can affect the number of genes with altered transcription [40,41,42,45]. For example, in *Populus trichocarpa* roots, more genes were up- and down-regulated at 2.5 μM TSA than at 1.0 μM TSA [32]. The results of the present study showed that *WUS* or *CYCD3-1* had different expression patterns depending on the incubation time and concentration of HDI. These results suggested that there are more HDI-induced changes in gene expression than previously reported.

Kruh (1982) reported that NaB (5 to 50 mM) caused the accumulation of acetylated histones H3 and H4 in mammalian cells [59]. In addition, the amount of acetylated histone proteins was increased in *Brassica napus* seedlings treated with 10 mM NaB for 2 weeks compared to that in control plants [60]. In addition, the treatment of tobacco seedlings for 6–24 h with 10 mM NaB or 10 μM TSA increased the acetylation state of histones H3 and H4 [61]. However, in alfalfa cell suspensions, the levels of acetylated histones declined 6 h after the addition of TSA, reaching levels similar to those in nontreated cells after 12 to 20 h [62]. The results of the present study showed that NaB increased histone acetylation in protoplast-derived calli of tobacco after 1 and 2 weeks of treatment compared to that at the beginning of treatment. These results suggested that NaB-induced acetylation in calli derived from protoplasts of tobacco persists for a long time.

The effect of NaB on adventitious shoot formation was different in tobacco and tomato cotyledon explants and protoplast derived calli. However, the effect of NaB on adventitious shoot formation in protoplast-derived tomato calli was not clear. In order to investigate whether NaB improves adventitious shoot formation in protoplast-derived tomato calli, the conditions (including hormones, carbon source, and culture medium) with high adventitious shoot formation efficiency should be established. In the case of 2,4-D, which is known to have the strongest potential for inducing somatic embryogenesis of plants so far, the ability to induce somatic cells differs significantly depending on the plant types or the explant types. This is a consequence of the differences in sensitivity to these hormones in each plant cell. As with plant growth regulators, it has been estimated that the sensitivity to NaB varies depending on the plant type or the tissues. However, in the present study, we could not determine why NaB had varying effects on cotyledon explants and tobacco protoplast-derived calli of tobacco despite their same origin (i.e., the cotyledons) and on shoot regeneration from calli derived from tomato and those derived from tomato protoplasts. Therefore, it is expected that the effect of NaB on adventitious shoot formation can be clarified by examining more the plant species in the future. We also investigated the acetylation of NaB in protoplast-derived tobacco calli; however; we did not assess the levels of histone acetylation in protoplast-derived tomato calli and cotyledon explants of tobacco and tomato. Thus, further studies are needed to determine if NaB acetylation occurs during adventitious shoot formation in protoplast-derived tomato calli and in tobacco and tomato cotyledon explants.

In summary, NaB significantly enhanced the growth and adventitious shoot formation of tobacco calli. However, NaB did not have a stimulating effect on shoot formation in tomato calli or in cotyledon explants of tobacco and tomato. These results suggest that the stimulating effect of NaB on adventitious shoot formation varies among the plant species and the explant types. In addition, the expression of *NbWUS* was accelerated by the NaB treatment in tobacco calli in a concentration-dependent manner, but this was delayed in tobacco explants in a concentration-dependent manner. Moreover, NaB enhanced the histone H3 acetylation level in tobacco calli in a concentration-dependent manner.

## 4. Materials and Methods

### 4.1. Plant Materials

Tobacco (*Nicotiana benthamiana*) seeds were sterilized with 70% (*v*/*v*) ethanol for 0.5 min and soaked in 0.8% sodium hypochlorite (NaOCl) solution for 0.5 min and then washed with sterile distilled water three times. Tomato (*Solanum lycopersicum* cv. Micro-Tom) seeds were sterilized with 70% (*v*/*v*) ethanol for 3 min and 0.8% sodium hypochlorite (NaOCl) solution for 3 min and then washed with sterile distilled water three times. The sterilized tomato seeds were kept in the dark for an initial 3-day period at 4 °C in water. The sterilized seeds were placed on Murashige and Skoog (MS) basal salt mixture (Duchefa, The Netherlands) supplemented with 3% (*w*/*v*) sucrose, 0.4 mg L^−1^ thiamine-HCl, 100 mg L^−1^ myo-inositol, and 0.4% (*w*/*v*) Gelrite, pH of the medium was adjusted to 5.8 with 1 N KOH, and the seeds were cultured in a growth chamber at 25 °C under a 16 h photoperiod (∼30 µmol m^−2^ s^−1^ cool white fluorescent light) and 70% relative humidity.

### 4.2. Protoplasts Isolation and Culture Condition

Cotyledons of 11-day-after germination (DAG11) tobacco seedlings were used as explants for protoplast isolation. Protoplasts were isolated as previously described [63] with slight modifications. Cotyledons were harvested; each cotyledon explant was cut into 1–2-mm-thick pieces and incubated with 10 mL of an enzyme solution [1% carbohydrase mix (Viscozyme; Novozymes, Denmark), 0.5% cellulose (Celluclast; Novozymes), 0.5% pectinase (Pectinex; Novozyme), 3 mM 2-(N-Morpholino) ethanesulfonic acid (MES), and 9% mannitol in CPW salts [64]] in a 100 mm × 20 mm plastic Petri dish on a gyratory shaker (50 rpm) at 25 °C for 5 h in the dark. After 5 h incubation, the protoplasts were passed through a 40 μm nylon cell strainer (Falcon, 352340) in a plastic Petri dish. The protoplasts were washed with W5 solution (2 mM MES, 154 mM NaCl, 125 mM CaCl_2_, and 5 mM KCl, pH 5.7 with 1 N KOH) three times (630 rpm for 5 min), after each wash, the supernatant was carefully removed. The obtained protoplasts pellet was suspended in initial protoplast culture medium (PCM) [MS medium containing 0.4 mg L^−1^ thiamine, 60 g L^−1^ myo-inositol, 3% (*w*/*v*) sucrose, 2 mg L^−1^ BAP, and 0.5 mg L^−1^ NAA; pH 5.8], and cell density was adjusted to 1 × 10^6^ cells mL^−1^ using a hemocytometer. 200 μL of each sample was immediately incubated in 1.8 mL of PCM at 25 °C in the dark for 3 weeks. After 3 weeks of incubation, micro-calli was transferred to 10 mL of PCM in a 100 mm × 20 mm plastic Petri dish on a gyratory shaker (70 rpm) at 25 °C under a 16 h photoperiod (∼30 µmol m^−2^ s^−1^ cool white fluorescent light) during 3–4 weeks.

Cotyledons of DAG7 tomato seedlings were used as explants for protoplast isolation. Cotyledon explants were harvested; each cotyledon explant was cut into 1–2-mm-thick pieces and incubated with 10 mL of an enzyme solution (described above) on a gyratory shaker (40 rpm) at 25 °C for 4.5 h in the dark. Protoplast isolation was carried out in the same manner as described above. After washing, the obtained protoplast pellet was suspended in PCM (MS medium containing 0.4 mg L^−1^ thiamine, 60 g L^−1^ myo-inositol, and 3% (*w*/*v*) sucrose, 2 mg L^−1^ NAA, and 0.5 mg L^−1^ Zeatin; pH 5.8).

### 4.3. The Effect of Sodium Butyrate in Protoplast-Derived Calli of Tobacco

To investigate the effect of sodium butyrate (NaB) on callus growth, callus size was measured in the callus induction medium (CIM) [B5 including B5 (Duchefa, Netherlands), 2% (*w*/*v*) sucrose, 2 mg L^−1^ BAP, 0.5 mg L^−1^ NAA, and 0.8% plant agar; pH 5.8] containing 0, 0.1, 0.5, and 1.0 mM of NaB after 5 weeks of incubation at 1-week intervals. Protoplast-derived calli of tobacco were transferred to the CIM and incubated for 2 weeks, after which the callus of 2 mm^2^ in size was transferred into 0, 0.1, 0.5, and 1.0 mM NaB-containing CIM.

Protoplast-derived calli of tobacco were transferred into the CIM and incubated for 4 weeks; thereafter, the callus of approximately 7 mm^2^ in size was transferred into the shoot induction medium (SIM) [1/2 MS basal salt mixture, 0.4 mg L^−1^ thiamine, 100 mg L^−1^ myo-inositol, 3% (*w*/*v*) sucrose, 2 mg L^−1^ IAA, 1 mg L^−1^ BA, and 0.8% plant agar; pH 5.8] containing 0, 0.1, 0.5, and 1.0 mM of NaB

The cultures were maintained under a 16 h photoperiod (∼30 µmol m^−2^ s^−1^ cool white fluorescent light) at 25 °C. Subculture of callus was carried out each month. Callus size was calculated by the program settled in Stereo Microscope (Nikon SMZ1270).

### 4.4. The Effect of Sodium Butyrate in Protoplast-Derived Calli of Tomato

After 3-4 weeks incubation in PCM, calli of tomato were transferred to the CIM [MS (Duchefa), 3% (*w*/*v*) sucrose, 0.4 mg L^−1^ thiamine, 100 mg L^−1^ myo-inositol, 2 mg L^−1^ NAA, 0.5 mg L^−1^ Zeatin, and 0.4% Gelite; pH 5.8] for 2–3 weeks, after which the callus of 2.5 mm^2^ in size was transferred into 0, 0.1, 0.5, and 1.0 mM NaB containing SIM [MS basal salt mixture (Duchefa), 3% (*w*/*v*) sucrose, 0.4 mg L^−1^ thiamine, 100 mg L^−1^ myo-inositol, 1 mg L^−1^ Zeatin and 0.1 mg L^−1^ IAA (SIM-1) or 2 mg L^−1^ Zeatin and 0.1 mg L^−1^ IAA (SIM-2), and 0.4% Gelite; pH 5.8]. The callus size was measured during 6 weeks of incubation at 1-week intervals in SIM-1 and SIM-2 containing 0, 0.1, 0.5, and 1.0 mM of NaB. The cultures were maintained under a 16 h photoperiod (∼30 µmol m^−2^ s^−1^ cool white florescent light) at 25 °C. Subculture of callus was performed each month. Callus size was calculated by the program settled in Stereo Microscope (Nikon SMZ1270).

### 4.5. Culture Conditions for Adventitious Shoot Formation in Cotyledon Explants of Tobacco and Tomato in the NaB Containing SIM

Cotyledons of DAG7 tomato seedlings were excised and divided into three segments, and cultured proximal segments in the SIM [MS basal salt medium (Duchefa), 3% (*w*/*v*) sucrose, 0.4 mg L^−1^ thiamine, 100 mg L^−1^ myo-inositol, 1 mg L^−1^ Zeatin, 0.1 mg L^−1^ IAA, and 0.4% (*w*/*v*) Gelrite; pH 5.8] containing 0, 0.1, 0.5, and 1.0 mM of NaB during 6 weeks of incubation.

Cotyledons of DAG11 tobacco seedlings were excised and cultured in the SIM [MS basal salt medium (Duchefa), 3% (*w*/*v*) sucrose, 0.4 mg L^−1^ thiamine, 100 mg L^−1^ myo-inositol, 2 mg L^−1^ BAP, 0.1 mg L^−1^ NAA, and 0.4% (*w*/*v*) Gelrite; pH 5.8)] containing 0, 0.1, 0.5, and 1.0 mM of NaB during 6 weeks of incubation.

All cultures were incubated under a 16 h photoperiod (∼30 µmol m^−2^ s^−1^ cool white fluorescent light) at 25 °C. The efficiency of adventitious shoots formation was calculated with the ratios from the total numbers of explants and the number of explants showing shoot formation.

### 4.6. Histone Extraction and Western Blotting

The calli cultured in the SIM containing 0, 0.1, 0.5, and 1.0 mM of NaB, respectively. After 1 or 2 weeks of incubation, calli were harvested and finely ground with pestle and bowl. Histone extraction performed using abcam’s protocol (https://www.abcam.com/protocols/histone-extraction-protocol-for-western-blot) with slight modifications. Histones were extracted by 0.2 N HCl overnight at 4 °C. Supernatant was collected and neutralized with 0.1 volume of 2 N NaOH in the supernatant. Histones were separated by 15% Sodium Dodecyl Sulfate Polyacrylamide Gel Electrophoresis (SDS-PAGE) and detected using a specific antibody against acetylated histone H3 (Merck Millipore, 06-599) and histone H3 antibody (Abcam, ab1791) by Chemiluminescence system Fusion Solo S (Vilver, France). The ratio of histone H3 and acetylated histone H3 was calculated using built-in software.

### 4.7. RNA Isolation and Quantitative Reverse Transcription PCR (qRT-PCR) Analysis

Cotyledon explants of tomato and tobacco or protoplast-derived calli of tobacco were incubated in the SIM containing NaB for 4, 8, and 12 days or 1, 2, and 3 weeks, respectively (as described above). Total RNA was extracted using an Aqueous Kit (Ambion) with a TURBO DNA-Free Kit (Applied Biosystems). cDNA was synthesized using a high-capacity cDNA Reverse Transcription Kit (Applied Biosystems). To investigate the transcript levels of *CYCD3-1* and *WUS* in individual tomato and tobacco explants, and protoplast-derived calli, each primer was designed using the Real-time PCR tool OligoAnalyzer 3.1 (Intergrated DNA Technologies, Coralville, IA, USA). The primer of *NbCYCD3-1* was designed to detect four of seven NbCYCD3-1 homologs (Niben101Scf05643g01012.1, Niben101Scf02445g10017.1, Niben101Scf01081g00003.1, and Niben101Scf04371g03001.1), and *NbWUS* gene information was obtained from Mlotshwa et al. (2006) [65]. The qRT-PCR was performed using an SYBR Green Kit (BioFACT, Korea) to detect the transcript levels of each gene on the CFX96^TM^ Real-Time PCR Detection System (Bio-Rad). The primers used for PCR were as follows: 5′-AAGCCCCTCTTAACCCAAAG-3′ and 5′-ACCAGAGTCCAACACAATACC-3′ for *SlActin4*; 5′-CAAGGAGAAGGTGGAAGGATG-3- and 5′-CTGTTGGACTCCCTGGTAATG-3′ for *SlCYCD3.1*; 5′-TGGAACTTTGGCTATGGAGAAG-3′ and 5′-GGGTAAGTTGCTGGAGAAGTAG-3′ for *SlWUS*; 5′-CCAAAGGCCAATCGAGAAAAG-3′ and 5′-CAGAGTCCAGCACAATACCAG-3′ for *NbActin*; 5′-CAACTTGTTGCTGTAGCTTGTC-3′ and 5′-CAAACACATATCTTGGATCCTCAAC-3′ for *NbCYCD3-1*; 5′-CATCCGCTTCTTCTCATGGT-3′ and 5′-GAGTTGCCACCTGGTGATATT-3′ for *NbWUS. SlActin4* (ACT4) and *NbActin* was used as an internal control.

Tomato gene sequence data from this article can be found in the Pyotozome v2.1 under the following accession numbers: *SlCYCD3-1* (Solyc04g078470.2), *SlWUSCHEL* (Solyc02g083950.2.1), *SlActin 4* (Solyc04g011500.2). *N. benthamiana* gene sequence data from this article can be found in the Sol genomics network and NCBI GenBank under the following accession numbers: *NbCYCD3-1* (Niben101Scf05643g01012.1), *NbACT* (Niben101Scf09133g02006.1), *NbWUS* (GenBank, DQ437634).

### 4.8. Statistical Analysis

Statistical analyses were carried out using values of experiments performed in triplicate. Data were analyzed using one-way analysis of variance test, followed by Tukey′s multiple comparisons using PRISM Software 8.0 (GraphPad Software, San Diego, CA, USA). Differences were considered statistically significant at *p* < 0.05.

## Figures and Tables

**Figure 1 ijms-21-08451-f001:**
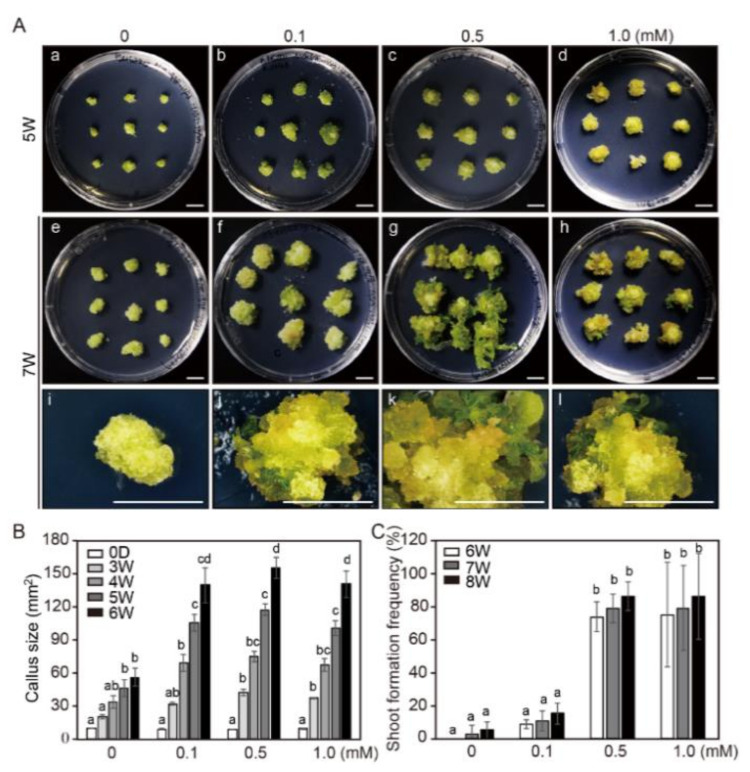
The effect of sodium butyrate (NaB) on adventitious shoot formation in protoplast-derived calli of tobacco. (**A**) Callus growth and adventitious shoot formation in the shoot induction medium (SIM) containing 0, 0.1, 0.5, and 1.0 mM of NaB after 5 (a–d) and 7 (e–l) weeks of culture. Scale bars = 1 cm. The size of callus (**B**) and the frequency of adventitious shoot formation (**C**) in the SIM containing 0, 0.1, 0.5, and 1.0 mM of NaB. Three independent experiments were performed on 81 callus. Error bars represent standard deviation (SD) (N = 81). Different letters on the bars indicate significant differences between each treatment (ANOVA followed by a Tukey’s test, *p* < 0.05). D = day; W = weeks.

**Figure 2 ijms-21-08451-f002:**
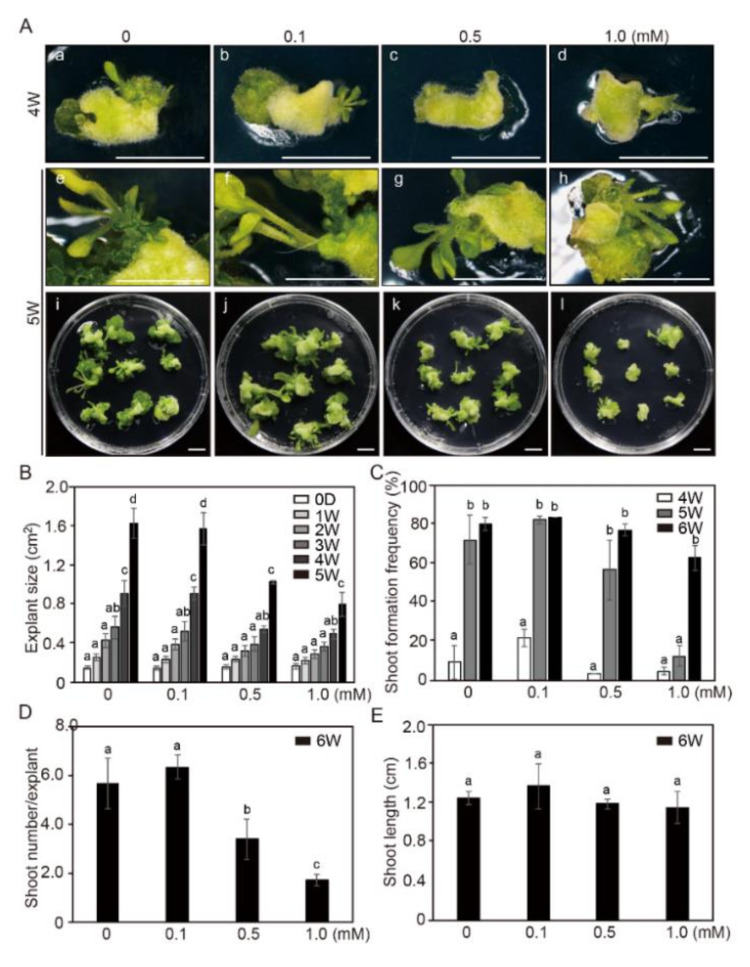
The effect of sodium butyrate (NaB) on adventitious shoot formation in cotyledon explants of tobacco. (**A**) Explant growth and adventitious shoot formation in the shoot induction medium (SIM) containing 0, 0.1, 0.5, and 1.0 mM of NaB after 4 (a–d) and 5 (e–l) weeks of culture. Scale bars = 1 cm. The size of cotyledon explants (**B**), the frequency of adventitious shoot formation (**C**), the shoot numbers per cotyledon explant (**D**), and shoot length (**E**) in the SIM containing 0, 0.1, 0.5, and 1.0 mM of NaB. Three independent experiments were performed on 81 explants. Error bars represent SD (N = 81). Different letters on the bars indicate significant differences between each treatment (ANOVA followed by a Tukey’s test, *p* < 0.05). D = day; W = week(s).

**Figure 3 ijms-21-08451-f003:**
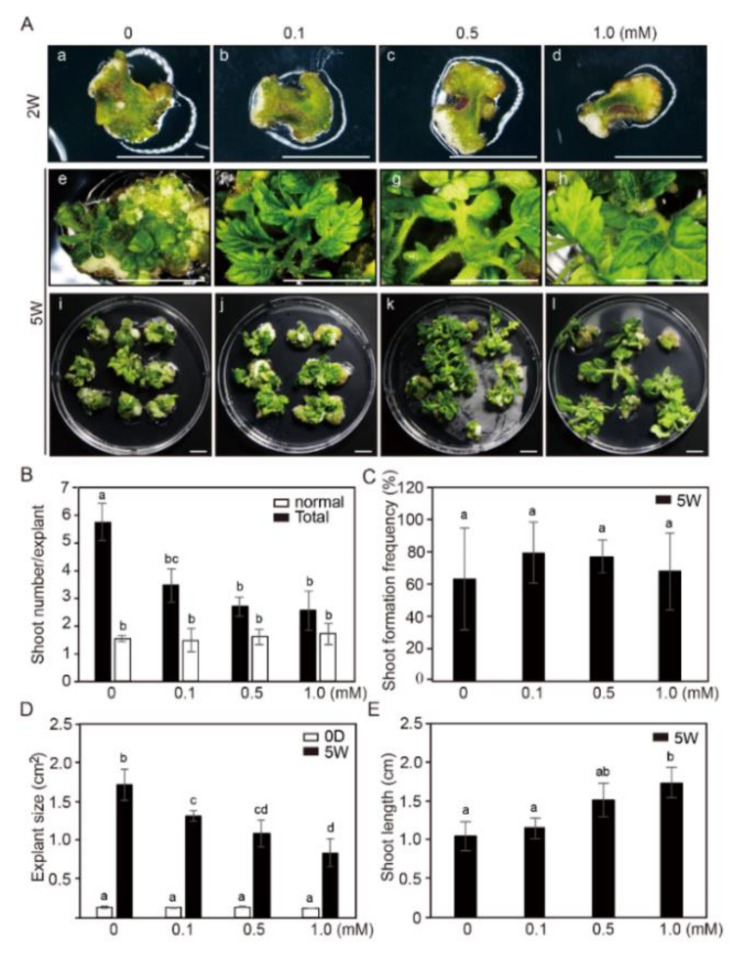
The effect of sodium butyrate (NaB) on adventitious shoot formation in cotyledon explants of tomato. (**A**) Explant growth and adventitious shoot formation in the shoot induction medium (SIM) containing 0, 0.1, 0.5, and 1.0 mM of NaB after 2 (a–d) and 5 (e–l) weeks of culture. Scale bars = 1 cm. The shoot numbers per cotyledon explant (**B**), the frequency of adventitious shoot formation (**C**), the size of tomato cotyledon explants (**D**), and the shoot length in cotyledon explants (**E**) in SIM-containing 0, 0.1, 0.5, and 1.0 mM of NaB. Three independent experiments were performed on 81 explants. Error bars represent SD (N = 81). Different letters on the bars indicate significant differences between each treatment (ANOVA followed by a Tukey’s test, *p* < 0.05). D = day; W = weeks.

**Figure 4 ijms-21-08451-f004:**
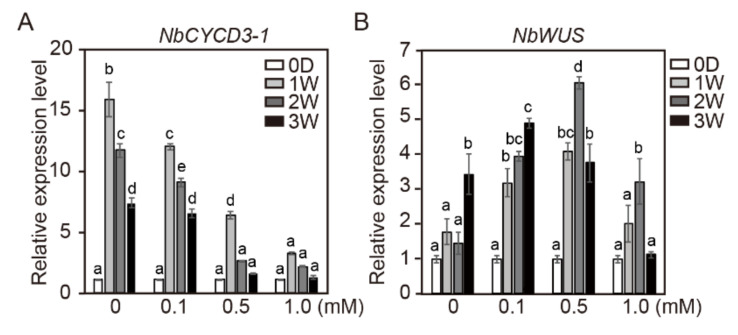
The relative expression levels of *NbCYCD3-1* and *NbWUS* genes in tobacco calli after NaB treatment. The relative gene expression level of *NbCYCD3-1* (**A**) and *NbWUS* (**B**) in tobacco calli cultured in shoot induction medium containing 0, 0.1, 0.5, and 1.0 mM of NaB. Data are representative of the results from three independent experiments. Error bars represent SD (*n* = 3). Different letters on the bars indicate significant differences between each treatment (ANOVA followed by a Turkey’s test, *p* < 0.05). D = day; W = week(s).

**Figure 5 ijms-21-08451-f005:**
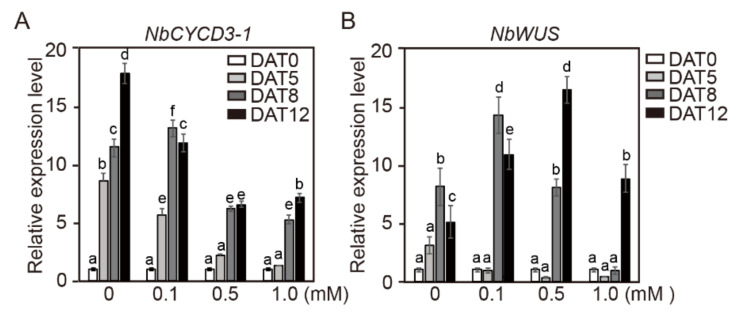
The relative expression levels of *NbCYCD3-1* and *NbWUS* genes in cotyledon explants of tobacco after NaB treatment. The relative gene expression level of *NbCYCD3-1* (**A**) and *NbWUS* (**B**) in cotyledon explants of tobacco cultured in shoot induction medium containing 0, 0.1, 0.5, and 1.0 mM of NaB. Data are representative of the results from three independent experiments. Error bars represent SD (*n* = 3). Different letters on the bars indicate significant differences between each treatment (ANOVA followed by a Turkey’s test, *p* < 0.05). DAT = day after treatment.

**Figure 6 ijms-21-08451-f006:**
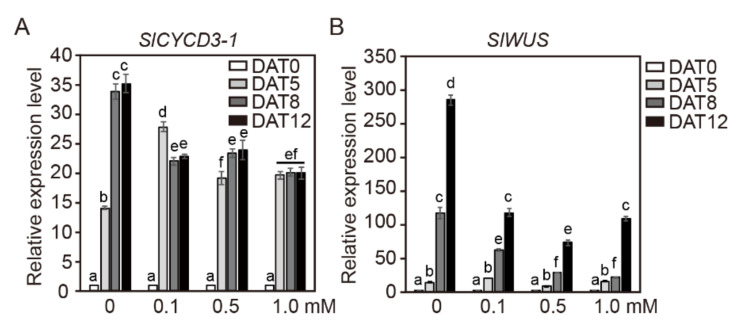
The relative expression levels of *SlCYCD3-1* and *SlWUS* genes in cotyledon explants of tomato after NaB treatment. The relative gene-expression levels of *SlCYCD3-1* (**A**) and *SlYUC10* (**B**) genes in cotyledon explants of tomato cultured in shoot induction medium containing 0, 0.1, 0.5, and 1.0 mM of NaB. Data are representative of the results from three independent experiments. Error bars represent SD (*n* = 3). Different letters on the bars indicate significant differences between each treatment (ANOVA followed by a Turkey’s test, *p* < 0.05). DAT = day after treatment.

**Figure 7 ijms-21-08451-f007:**
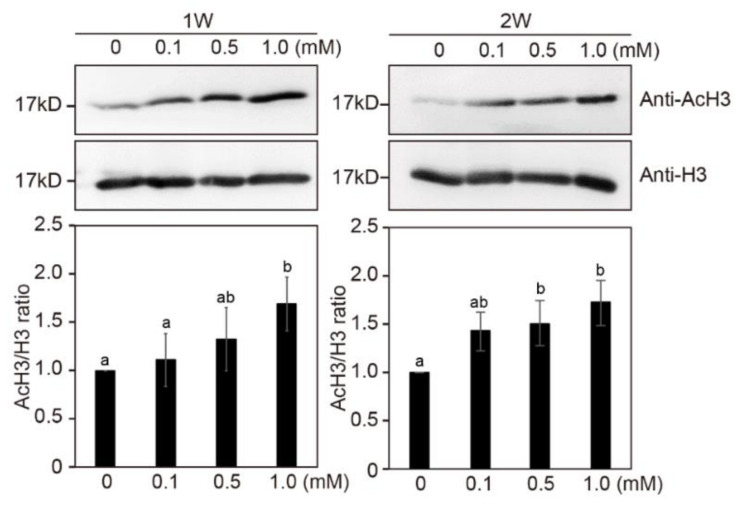
The effect of sodium butyrate on histone H3 acetylation level in tobacco protoplast-derived calli. Total protein extracts were obtained from protoplast-derived calli of tobacco after 1 or 2 weeks after NaB treatments. Error bars represent SD (*n* = 3). Histone acetylation levels were determined with a western blot using an anti-H3 and anti-AcH3 antibody. Different letters on the bars indicate significant differences between each treatment (ANOVA followed by a Turkey’s test, *p* < 0.05). W = week(s); H3 = histone H3 antibody; AcH3 = acetylated histone H3 antibody.

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
