# Peer review of "The Effect of Sodium Butyrate on Adventitious Shoot Formation Varies among the Plant Species and the Explant Types"

_ijms, 2020, doi:10.3390/ijms21228451_

Round 1

Reviewer 1 Report

Authors investigated the effect of sodium butyrate (NaB), a histone deacetylase inhibitor, on adventitious shoot formation from protoplast-derived calli and cotyledon explants of tobacco (Nicotiana benthamiana) and tomato (Solanum lycopersicum).Results were well presented, the experimental design is robust, including control and a considerable number of biological replicates.  The effects of NaB on calli from protoplast of tobacco are clear, but effects on tomato calli were less impressive. Discussion on those data seems to be in accordance and come to complete the comprehension of results.  The paper could be accepted with some modification.

Some minor changes may to be required:

Abstract: First sentence (line 12) needs to be extended to indicate that NaB is an inhibitor of this kind of process, because it is not clear on the abstract  

In paragraph 51-59 author indicated that NaB has been reported to induce a wide variety of plant physiological responses, such as INHIBITION of cell division. However in paragraph 60-66, authors propose to study the effect of effects of NaB on the ACCELERATION of cell division from protoplasts and adventitious shoot formation. Are not those phenomena opposite between them? How to explain?

Paragraph 63 – remove FROM

Section 2.4. Authors talk about normal and abnormal shot growth. Could authors be more precise about what a shoot coming from protoplasts can be defined as normal or not?

Paragraoh 184. Authors propose to study expression of two genes related with shoot cell differentiation, CYCD-3 and WUS, because expression of those genes could explain phenotypes observed by NaB treatment, and so be deacetylated and downregulated. Can authors explain the choice of those two genes in term of their acetylation status during cell division? There is a way to quantify acetylation status of promotors of those genes before and after NaB treatment? By example a ChIP-histone PCR? If not, author may include in the discussion some futures experiments to address that question.

Best.

Author Response

Authors investigated the effect of sodium butyrate (NaB), a histone deacetylase inhibitor, on adventitious shoot formation from protoplast-derived calli and cotyledon explants of tobacco (Nicotiana benthamiana) and tomato (Solanum lycopersicum).Results were well presented, the experimental design is robust, including control and a considerable number of biological replicates.  The effects of NaB on calli from protoplast of tobacco are clear, but effects on tomato calli were less impressive. Discussion on those data seems to be in accordance and come to complete the comprehension of results. The paper could be accepted with some modification.

Some minor changes may to be required:

Abstract: First sentence (line 12) needs to be extended to indicate that NaB is an inhibitor of this kind of process, because it is not clear on the abstract

Response:

Thanks for your comments.

In abstract section, the role of NaB in histone acetylation was added as ‘a histone deacetylase inhibitor’.

In paragraph 51-59 author indicated that NaB has been reported to induce a wide variety of plant physiological responses, such as INHIBITION of cell division. However in paragraph 60-66, authors propose to study the effect of effects of NaB on the ACCELERATION of cell division from protoplasts and adventitious shoot formation. Are not those phenomena opposite between them? How to explain?

Response:

We thank the reviewer for bring this important point.

To eliminate unintended confusion, we changed ‘inhibition of cell division’ to cell division in explants’ and deleted ‘acceleration of cell division from protoplasts and’. In fact, our study cannot explain the accelerated cell division from protoplast given the various effects of NaB on adventitious shoot formation of tomatoes and tobacco, or the various results on gene expression of cell cyclin D3-1.

Paragraph 63 – remove FROM

Response:

We removed the word ‘from’.

Section 2.4. Authors talk about normal and abnormal shot growth. Could authors be more precise about what a shoot coming from protoplasts can be defined as normal or not?

Response:

Thanks for your comment.

In fact, a normal or abnormal shoot is a subjective expression, but explanation was insufficient. We modified ‘morphologically normal shoot’ to ‘morphologically normal shoot formation (excluding abnormal-shaped shoots, such as those with a single leaf)’, and abnormal shoot’ to ‘abnormal-shaped shoots (such as examples with a single leaf)’.

Paragraoh 184. Authors propose to study expression of two genes related with shoot cell differentiation, CYCD-3 and WUS, because expression of those genes could explain phenotypes observed by NaB treatment, and so be deacetylated and downregulated. Can authors explain the choice of those two genes in term of their acetylation status during cell division? There is a way to quantify acetylation status of promotors of those genes before and after NaB treatment? By example a ChIP-histone PCR? If not, author may include in the discussion some futures experiments to address that question.

Response:

The reasons for choosing CYCD3-1 and WUS were added to the introduction section and results 2.5 section.

In introduction section:

‘In Arabidopsis, the genes such as CUPSHAPED COTYLEDON (CUC), SHOOT MERISTEMLESS (STM), WUSCHEL (WUS), and many others are required for shoot apical meristem (SAM) formation during embryogenesis for SAM maintenance during post‐embryonic development [5-8]. CUC1 and CUC2, encoding a pair of paralogous NAC transcription factors, are required for shoot meristem initiation through the promotion of STM expression [9]. STM promotes cell division and inhibits cell differentiation in the SAM [10]. WUS, a homeodomain transcription factor, is essential for reprogramming during de novo shoot regeneration [11]. One study showed that a wus mutant failed to regenerate the shoots, whereas the overexpression of WUS led to shoot regeneration on a hormone- free medium [12]. Type-B Arabidopsis response regulators (ARRs) play an important role in the regulation of auxin levels by cytokinin signaling [13], which leads to cell-cycle re-entry via the up-regulation of D-type cyclin CYCD3;1 [14]. CYCD3;1 functions as a downstream effector of cytokinin signaling, and it is involved in the control of the cell cycle at the G1/S transition [15].’

‘In Brassica napus, TSA treatment increased histone H3 and H4 acetylation of the WUS genomic region [24] and upregulated a few members of cell cycle-related genes in the male gametophyte [25]. In addition, cell cycle-related genes were up- and down-regulated by many HDI treatments [26].’

In result 2.5 section:

‘Histone modification is widely associated with the transcriptional regulation of cell cycle genes [48]. Li et al. (2014) reported that a small number of cell cycle-related genes (cyclin D3;3, CYCLIN D1-like gene) were upregulated after TSA treatment [25]. TSA treatment enabled rapid de novo activation of WUS expression by cytokinin analog BAP [24]. In addition, the expression of the M/G2-phase marker CYCB1;1 in callus was higher in hag1 (histone acetyltransferase) mutant than in wild type plants, whereas WUS expression was dispersed in hag1 callus cultured on SIM and defected shoot formation [49].’

Reviewer 2 Report

This manuscript has evaluated the role of histone deacetylase inhibitors (HDI), NaB, in adventitious shoot formation of tobacco and tomato tissues, showing various phanotypic and gene expression response, which provides insights into plant shoot-regeneration. Overall, the experiment design and data analysis are reasonable, the writing and result description are clear. I only have some minor comments, including conclusion of gene expression and data explanation.

  1. In Abstract, the authors should introduce the role of NaB in histone acetylation

  1. Line 118, ‘Figure 1A, B’ should be Figure ‘2A,B’

  1. Comments on Line 136-138, The different effects of NaB concentrations on cotyledon looks like the opposite functions of some plant hormones in various plant tissues, such as auxin. Did the author test the lower NaB concentrations from 0.01-0.1 mM?

  1. How to link the CYCD3-1 and WUS expression in adventitious shoot formation? Is there any genetic data that have been reported, like mutants?

  1. In Figure 4A, the expression level of CYCD3-1 decreased significantly with the increase in NaB concentration, how the author can draw the conclusion in Line 203-204 'These results suggested that the expression of CYCD3-1 and WUS in tobacco calli was more rapidly induced and then reduced by the NaB treatment during the culture in SIM.'

  1. In Result 2.6, even the delayed expression of CYCD3-1 and WUS in cotyledon explants, compared with untreated control, the expression of CYCD3-1 was repressed and WUS was promoted, which is similar with that of Calli. Based on the different effects of NaB in these two tissues, how to explain this variation?

  1. In Result 2.8. Why only to show the histone acetylation in tobacco protoplast-derived calli? how about in cotyledon explants of tobacco and tomato samples?

  1. This study demonstrated the various responses of NaB treatments in different tissues and species (tobacco and tomato), but the explanations of this variation looks insufficient. The authors should at least discuss this point at molecular and physiological levels or tissue/species-specific response.

Author Response

This manuscript has evaluated the role of histone deacetylase inhibitors (HDI), NaB, in adventitious shoot formation of tobacco and tomato tissues, showing various phanotypic and gene expression response, which provides insights into plant shoot-regeneration. Overall, the experiment design and data analysis are reasonable, the writing and result description are clear. I only have some minor comments, including conclusion of gene expression and data explanation.

  1. In Abstract, the authors should introduce the role of NaB in histone acetylation

Response:

In Abstract section, the role of NaB in histone acetylation is introduced as ‘a histone deacetylase inhibitor’.

  1. Line 118, ‘Figure 1A, B’ should be Figure ‘2A,B’

Response:

Thanks for pointing out the mistake, we corrected ‘Figure 1A,B’ to ‘Figure 2A,B’.

  1. Comments on Line 136-138, The different effects of NaB concentrations on cotyledon looks like the opposite functions of some plant hormones in various plant tissues, such as auxin. Did the author test the lower NaB concentrations from 0.01-0.1 mM?

Response:

We tested lower NaB concentration of 0.5-2 ppm (0.45-1.8 mM) in tomato explants, but showed a similar phenotype to the control. Therefore, concentrations lower than 0.1 mM was not included in other experiments and the effect by NaB was only investigated at 0.1-1.0 mM.

  1. How to link the CYCD3-1 and WUS expression in adventitious shoot formation? Is there any genetic data that have been reported, like mutants.

Response:

The reasons for choosing CYCD3-1 and WUS were added to the introduction and results 2.5 section.

In introduction section:

‘In Arabidopsis, the genes such as CUPSHAPED COTYLEDON (CUC), SHOOT MERISTEMLESS (STM), WUSCHEL (WUS), and many others are required for shoot apical meristem (SAM) formation during embryogenesis for SAM maintenance during post‐embryonic development [5-8]. CUC1 and CUC2, encoding a pair of paralogous NAC transcription factors, are required for shoot meristem initiation through the promotion of STM expression [9]. STM promotes cell division and inhibits cell differentiation in the SAM [10]. WUS, a homeodomain transcription factor, is essential for reprogramming during de novo shoot regeneration [11]. One study showed that a wus mutant failed to regenerate the shoots, whereas the overexpression of WUS led to shoot regeneration on a hormone- free medium [12]. Type-B Arabidopsis response regulators (ARRs) play an important role in the regulation of auxin levels by cytokinin signaling [13], which leads to cell-cycle re-entry via the up-regulation of D-type cyclin CYCD3;1 [14]. CYCD3;1 functions as a downstream effector of cytokinin signaling, and it is involved in the control of the cell cycle at the G1/S transition [15].’

“In Brassica napus, TSA treatment increased histone H3 and H4 acetylation of the WUS genomic region [24] and upregulated a few members of cell cycle-related genes in the male gametophyte [25]. In addition, cell cycle-related genes were up- and down-regulated by many HDI treatments [26]”

In result 2.5 section:

‘Histone modification is widely associated with the transcriptional regulation of cell cycle genes [48]. Li et al. (2014) reported that a small number of cell cycle-related genes (cyclin D3;3, CYCLIN D1-like gene) were upregulated after TSA treatment [25]. TSA treatment enabled rapid de novo activation of WUS expression by cytokinin analog BAP [24]. In addition, the expression of the M/G2-phase marker CYCB1;1 in callus was higher in hag1 (histone acetyltransferase) mutant than in wild type plants, whereas WUS expression was dispersed in hag1 callus cultured on SIM and defected shoot formation [49].’

  1. In Figure 4A, the expression level of CYCD3-1 decreased significantly with the increase in NaB concentration, how the author can draw the conclusion in Line 203-204 'These results suggested that the expression of CYCD3-1 and WUS in tobacco calli was more rapidly induced and then reduced by the NaB treatment during the culture in SIM.'

Response:

The expression of NbCYCD3-1 decreased significantly with increasing NaB concentration. We speculated that, as with NbWUS, the expression of NbCYCD3-1 would increase faster than the control earlier than one week of NaB treatment and then decrease again. However, in the study, it is not clear whether the expression of NbCYCD3-1 increased faster than the control earlier than one week by NaB treatment and then decreased again compared to the control. Therefore, after receiving your advice, we further compared the expression patterns of NbCYCD3-1 in protoplast-derived tobacco calli by qPCR at 2, 4, and 7 days after NaB treatment. It had a similar pattern to 1-3 weeks after treatment, and was different from the expression pattern of NbWUS (we did not include additional experimental data in the manuscript). Therefore, I have modified the manuscript (In Abstract, Results 2.5, and Discussion) following your advice.

à Result 2.5 section:

‘These results suggested that the expression of NbWUS in tobacco calli was more rapidly induced and then reduced by the NaB treatment during the culture in SIM. Furthermore, these results also indicated that the effect of NaB on the early expression of WUS varies depending on the NaB concentration. However, the expression of NbCYCD3-1 decreased significantly with increasing NaB concentration.’

  • In Result 2.6, even the delayed expression of CYCD3-1 and WUS in cotyledon explants, compared with untreated control, the expression of CYCD3-1 was repressed and WUS was promoted, which is similar with that of Calli. Based on the different effects of NaB in these two tissues, how to explain this variation?

Response:

In fact, the adventitious shoot formation in NaB and the expression patterns of CYCD3-1 and WUS vary with explants in callus and plant species. At molecular biological level, we cannot give a sufficient answer because there are not enough data to explain this phenomenon. Instead, the opinions below regarding this have been e included in the discussion section.

‘In the case of 2,4-D, which is known to have the strongest potential for inducing somatic embryogenesis of plants so far, the ability to induce somatic cells differs significantly depending on the plant types or the explant types . This is a consequence of the differences in sensitivity to these hormones in each plant cell. As with plant growth regulators, it has been estimated that the sensitivity to NaB varies depending on the plant type or the tissues. However, in the present study, we could not determine why NaB had varying effects on cotyledon explants and tobacco protoplast-derived calli of tobacco despite their same origin (i.e., the cotyledons) and on shoot regeneration from calli derived from tomato and those derived from tomato protoplasts. Therefore, it is expected that the effect of NaB on adventitious shoot formation can be clarified by examining in more the plant species in the future.’

  1. In Result 2.8. Why only to show the histone acetylation in tobacco protoplast-derived calli? how about in cotyledon explants of tobacco and tomato samples?

Response:

Protoplast-derived calli of tobacco showed a positive effect on the adventitious shoots formation, so we observed the effect of NaB treatment in tobacco calli. However, we did not performed experiments observing the effects of NaB in cotyledon explants of tobacco and tomatoes. In fact, we performed a similar experiment on protoplast-derived calli of cabbage. In the cotyledon explants of cabbage, the adventitious shoot formations had a negative effect on callus growth in shoot induction medium. After 1 and 2 weeks of incubation in shoot induction medium containing NaB, histone H3 acetylation in cabbage calli was similar to that of the control. However, to observe change in the level of histone acetylation by NaB in tobacco and tomato explants, additional experiments with incubation time and concentration are required .

Therefore, we included the following sentence to the discussion section.

‘Thus, further studies are needed to determine if NaB acetylation occurs during adventitious shoot formation in protoplast-derived tomato calli and in tobacco and tomato cotyledon explants.’

  1. This study demonstrated the various responses of NaB treatments in different tissues and species (tobacco and tomato), but the explanations of this variation looks insufficient. The authors should at least discuss this point at molecular and physiological levels or tissue/species-specific response.

Response:

As in response to comment 6, we don’t have enough data and reference to explain this phenomenon at the molecular biological level. I think further studies should be conducted. Like the answer to comment 6, I have added a few sentences to the discussion section.

Reviewer 3 Report

Comments and suggestions for the authors:

MS# ijms-974598

A study concerning the “effect of sodium butyrate (NaB) on adventitious shoot formation from protoplast-derived calli and cotyledon explants of tobacco and tomato” has been carried out. The presentation of findings and data interpretation in this manuscript could have been improved compared to this current format. However, the overall comments /or opinions concerning the manuscript are as follows:

Major Comments:                                                                                                                

  1. The author should be checked the language through the manuscript via Native Speaker with a similar field of experience.

  1. Gene functions of CYCD3 and WES should be clearly defined in introduction section with relevant references.

  1. Discussion section should be concised. A good discussion includes: (a) principal, relationship and generalizations that can be supported by the results. (b) Emphasize findings and assumptions that

support or disagree with other work(s). (c) Exceptions, lack of correlations, gap area needing further investigation, and conclusion. Note, repetition of the results should be avoided in discussion section.

  1. The study gap should be correctly defined in the introduction section.

Minor Comments:

-Line 3, page 1: “the” should be added after the world “among” and “and” in title.

- Lines 32-37, page 1: should be revised with simple present tense.

- After the line 59, page 2: one small section should be added related to proper functions of CYCD3 and WES with references.

-Research gap should be raised properly in introduction section.

- After the line 118, page 4: result should be added in support of Fig. 1C. Subsequently, results of 2A should be inserted.

-Lines 182-185, page 7: Clear biological function of CYCD3 and WES genes should be added.

-Lines 225-227: CYCD3 and WES gene functions are quict different. So, how the author confirmed these genes together significantly delayed by the NaB. Please check and revise the lines 225-227.

-Page 9: title of Fig. 6: Gene name “SICYCD3-1” and “SIWUS” should be replaced by “SlCYCD3-1” “SlWUS”

-

-Lines 227, 300: page 10: value with results should be avoided in discussion section

- Lines 316, 317,320,337, page 11: it is better to avoid figure number especially in discussion section.

- Page 12-15: Numbering of subsection (4.1~4.8) should be maintained in materials and methods section

- Lines 466-475, page 14: should be concised.

-Line 475, page 14: The word “prepared” should be replaced by the word “Extracted”

-Page 15, Statistical analysis: how many replications were maintained? The author should be stated clearly.

Author Response

A study concerning the “effect of sodium butyrate (NaB) on adventitious shoot formation from protoplast-derived calli and cotyledon explants of tobacco and tomato” has been carried out. The presentation of findings and data interpretation in this manuscript could have been improved compared to this current format. However, the overall comments /or opinions concerning the manuscript are as follows:

Major Comments:                                                                                                                

  1. The author should be checked the language through the manuscript via Native Speaker with a similar field of experience.

Response:

In fact, our manuscript received English editing services. In addition, we have been re-edited after revising the manuscripts according to your advice.

  1. Gene functions of CYCD3 and WES should be clearly defined in introduction section with relevant references.

Response:

Thanks for your comments. In introduction section, we have defined the CYCD3 and WUS as below.

‘In Arabidopsis, the genes such as CUPSHAPED COTYLEDON (CUC), SHOOT MERISTEMLESS (STM), WUSCHEL (WUS), and many others are required for shoot apical meristem (SAM) formation during embryogenesis for SAM maintenance during post‐embryonic development [5-8]. CUC1 and CUC2, encoding a pair of paralogous NAC transcription factors, are required for shoot meristem initiation through the promotion of STM expression [9]. STM promotes cell division and inhibits cell differentiation in the SAM [10]. WUS, a homeodomain transcription factor, is essential for reprogramming during de novo shoot regeneration [11]. One study showed that a wus mutant failed to regenerate the shoots, whereas the overexpression of WUS led to shoot regeneration on a hormone- free medium [12]. Type-B Arabidopsis response regulators (ARRs) play an important role in the regulation of auxin levels by cytokinin signaling [13], which leads to cell-cycle re-entry via the up-regulation of D-type cyclin CYCD3;1 [14]. CYCD3;1 functions as a downstream effector of cytokinin signaling, and it is involved in the control of the cell cycle at the G1/S transition [15].’

  1. Discussion section should be concised. A good discussion includes: (a) principal, relationship and generalizations that can be supported by the results. (b) Emphasize findings and assumptions that

support or disagree with other work(s). (c) Exceptions, lack of correlations, gap area needing further investigation, and conclusion. Note, repetition of the results should be avoided in discussion section.

Response:

We have rewritten the discussion section. Similar sentence, value with results, and figure number were deleted and the order of the sentences was changed. In addition, gap area that need further investigation are included before the summary.

  1. The study gap should be correctly defined in the introduction section.

Response:

We included a description of the role of CYCD3-1 and WUS and the function of these two genes in adventitious shoot formation in the introduction section. In addition, in the introduction and results sections 2.5 and 2.8 added the link between these two genes in histone acetylation and reasons for investigating the gene expression of CYCD3-1 and WUS after NaB treatment.

Minor Comments:

-Line 3, page 1: “the” should be added after the world “among” and “and” in title.

Response:

We added ‘the’ in title and in discussion section.

- Lines 32-37, page 1: should be revised with simple present tense.

Response:

The sentence has been changed as below.

‘The ability of plants to dedifferentiate and regenerate cells from differentiated somatic tissues is plants has served as an important tool for producing fully functional plantlets from plant explants or cells. The regenerative capacity of plant cells can be enhanced in vitro in nutrient media supplemented with plant hormones [1,2], and in particular, the ratio of auxin to cytokinin plays an important role in the determination of shoot, root, or callus differentiation [1,3].’

And the following sentence was added.

‘Callus formation from explants can be induced on auxin-rich callus-inducing medium (CIM) and shoot regeneration from callus cells can be induced on cytokine-rich shoot induction medium (SIM) [4].’

- After the line 59, page 2: one small section should be added related to proper functions of CYCD3 and WES with references.

Response:

We included a description of the role of CYCD3-1 and WUS and the function of these two genes in adventitious shoot formation in the introduction section.

-Research gap should be raised properly in introduction section.

We added related to proper functions of CYCD3 and WES with references in lines 40-63 as below.

‘In Arabidopsis, the genes such as CUPSHAPED COTYLEDON (CUC), SHOOT MERISTEMLESS (STM), WUSCHEL (WUS), and many others are required for shoot apical meristem (SAM) formation during embryogenesis for SAM maintenance during post‐embryonic development [5-8]. CUC1 and CUC2, encoding a pair of paralogous NAC transcription factors, are required for shoot meristem initiation through the promotion of STM expression [9]. STM promotes cell division and inhibits cell differentiation in the SAM [10]. WUS, a homeodomain transcription factor, is essential for reprogramming during de novo shoot regeneration [11]. One study showed that a wus mutant failed to regenerate the shoots, whereas the overexpression of WUS led to shoot regeneration on a hormone- free medium [12]. Type-B Arabidopsis response regulators (ARRs) play an important role in the regulation of auxin levels by cytokinin signaling [13], which leads to cell-cycle re-entry via the up-regulation of D-type cyclin CYCD3;1 [14]. CYCD3;1 functions as a downstream effector of cytokinin signaling, and it is involved in the control of the cell cycle at the G1/S transition [15]./

And also added an additional sentence in lines 74-77 as below.

‘In Brassica napus, TSA treatment increased histone H3 and H4 acetylation of the WUS genomic region [24] and upregulated a few members of cell cycle-related genes in the male gametophyte [25]. In addition, cell cycle-related genes were up- and down-regulated by many HDI treatments [26].’

- After the line 118, page 4: result should be added in support of Fig. 1C. Subsequently, results of 2A should be inserted.

Response:

We replaced Figure 1 to Figure 1C and modified Figure 1A, B to Figure 2A,B in results section 2.3.

-Lines 182-185, page 7: Clear biological function of CYCD3 and WES genes should be added.

Response:

We added biological function of CYCD3 and WES genes in introduction section.

-Lines 225-227: CYCD3 and WES gene functions are quict different. So, how the author confirmed these genes together significantly delayed by the NaB. Please check and revise the lines 225-227.

Response”

We revised the sentence as follows.

‘These results suggested that the expression of both NbCYCD3-1 and NbWUS in cotyledon explants was negatively affected by the NaB treatment compared to that in the control.’

-Page 9: title of Fig. 6: Gene name “SICYCD3-1” and “SIWUS” should be replaced by “SlCYCD3-1” “SlWUS”

Response:

Your comment is confusing. However, in the manuscript, tomato CYCD3-1 and WUS were rewritten as SlCYCD3-1 and SlWUS, and tobacco CYCD3-1 and WUS were rewritten as NbCYCD3-1 and NbWUS.

-Lines 227, 300: page 10: value with results should be avoided in discussion section

Response:

We deleted line 292-302 in discussion section.

- Lines 316, 317,320,337, page 11: it is better to avoid figure number especially in discussion section.

Response:

We deleted figure numbers in discussion section.

- Page 12-15: Numbering of subsection (4.1~4.8) should be maintained in materials and methods section

Response:

We added numbering of subsection.

- Lines 466-475, page 14: should be concised.

Response:

lines 466-475 have been consisted as below.

“Cotyledon explants of tomato and tobacco or protoplast-derived calli of tobacco were incubated in the SIM containing NaB for 4, 8, and 12 days or 1, 2, and 3 weeks, respectively (as described above).”

-Line 475, page 14: The word “prepared” should be replaced by the word “Extracted”

Response:

We replaced “prepared” to “extracted.

-Page 15, Statistical analysis: how many replications were maintained? The author should be stated clearly.

Response:

We revised as below.

‘Statistical analyses were carried out using values of experiments performed in triplicate.’

Round 2

Reviewer 3 Report

Dear Authors,

MS#ijms-974598

Thank you for the point-by-point revision of the manuscript. The built-in improvements render this manuscript better than before.

Please try to maintain a consequence of Figure numbers, as two plants have been considered. In case of supplementary figure, it is better to mention like as supplementary figure S1 or supplementary figure S1A.This manuscript may be considered for further processing after the completion of the minor revisions without a track change mode with word file.

Regards,

Anonymous reviewer

Author Response

Please try to maintain a consequence of Figure numbers, as two plants have been considered. In case of supplementary figure, it is better to mention like as supplementary figure S1 or supplementary figure S1A.This manuscript may be considered for further processing after the completion of the minor revisions without a track change mode with word file.

Response:

Thanks for your comments.

We modified 'Figure S1-S3' to 'Supplementary Figure S1-S3' in the Results section, Supplementary materials section, and Supplementary figure captions.